# *Viscum album* Induces Apoptosis by Regulating STAT3 Signaling Pathway in Breast Cancer Cells

**DOI:** 10.3390/ijms241511988

**Published:** 2023-07-26

**Authors:** Ye-Rin Park, Wona Jee, So-Mi Park, Seok Woo Kim, Hanbit Bae, Ji Hoon Jung, Hyungsuk Kim, Sangki Kim, Jong Sup Chung, Hyeung-Jin Jang

**Affiliations:** 1College of Korean Medicine, Kyung Hee University, 24, Kyungheedae-ro, Dongdaemun-gu, Seoul 02447, Republic of Korea; yerinp@khu.ac.kr (Y.-R.P.); psm1030@naver.com (W.J.); psm991030@naver.com (S.-M.P.); kim66470@naver.com (S.W.K.); gksqlc4321@naver.com (H.B.); johnsperfume@gmail.com (J.H.J.); kim0874@hanmail.net (H.K.); 2Department of Science in Korean Medicine, Graduate School, Kyung Hee University, Seoul 02447, Republic of Korea; 3Department of Korean Rehabilitation Medicine, Kyung Hee University Medical Center, Seoul 02447, Republic of Korea; 4Dalim Biotech, 33 Sinpyeong-ro, Jijeong-myeon, Wonju-si 26348, Republic of Korea; plan713@dalimpharm.co.kr (S.K.); cjsup47@naver.com (J.S.C.)

**Keywords:** *Viscum album*, STAT3, SHP-1, breast cancer, doxorubicin

## Abstract

In this study, we investigated the potential anticancer effects of *Viscum album*, a parasitic plant that grows on *Malus domestica* (VaM) on breast cancer cells, and explored the underlying mechanisms. VaM significantly inhibited cell viability and proliferation and induced apoptosis in a dose-dependent manner. VaM also regulated cell cycle progression and effectively inhibited activation of the STAT3 signaling pathway through SHP-1. Combining VaM with low-dose doxorubicin produced a synergistic effect, highlighting its potential as a promising therapeutic. In vivo, VaM administration inhibited tumor growth and modulated key molecular markers associated with breast cancer progression. Overall, our findings provide strong evidence for the therapeutic potential of VaM in breast cancer treatment and support further studies exploring clinical applications.

## 1. Introduction

Breast cancer is the most common malignancy affecting women worldwide and accounts for many cancer-related deaths [1,2]. Despite advances in early detection and therapeutic interventions, the development of resistance to conventional treatments and the high risk of recurrence remain major challenges in the management of breast cancer [3,4,5]. Therefore, there is a pressing need to explore alternative treatment strategies to target breast cancer cells and overcome drug resistance effectively.

Apoptosis, also known as programmed cell death, plays a critical role in maintaining tissue homeostasis and eliminating damaged or cancerous cells [6,7,8]. Dysregulation of apoptosis is a hallmark of cancer, and strategies that selectively induce apoptosis in cancer cells have significant therapeutic potential [9,10].

Cell cycle dysregulation is another hallmark of cancer, and targeting cell cycle progression has emerged as a promising therapeutic approach [11,12]. Cyclins and cyclin-dependent kinases are key regulators of cell cycle progression, and aberrant expression of these molecules is commonly observed in cancer cells [13,14,15].

The signal transducer and activator of transcription 3 (STAT3) is a transcription factor that plays a critical role in promoting the survival, proliferation, and immune evasion of cancer cells [16,17]. Aberrant activation of STAT3 has been observed in many cases of breast cancer and is associated with poor prognosis and resistance to conventional therapies [18,19,20]. Therefore, targeting the STAT3 pathway is an attractive strategy for breast cancer treatment.

Natural products have gained increasing attention as potential sources of novel anticancer agents [21,22]. The extract used in this study is derived from *Viscum album*, a parasitic plant that grows on *Malus domestica* (VaM) and has been used in complementary and alternative medicine for centuries due to its potential anticancer properties [23,24,25,26,27]. However, the effect of VaM on STAT3 signaling in breast cancer cells has not been extensively studied.

Combination therapy approaches have gained increasing attention in cancer treatment because of their potential to enhance efficacy and overcome drug resistance [28]. Therefore, exploring the potential synergistic effects of VaM with standard chemotherapeutic agents, such as doxorubicin, in breast cancer cells could provide valuable insights into novel combination therapy strategies.

In this study, we investigated the anticancer effects of VaM on breast cancer cells and elucidated its underlying molecular mechanisms. Specifically, we examined the effect of VaM on cell viability, proliferation, apoptosis induction, cell cycle progression, and regulation of the STAT3 signaling pathway in breast cancer cells. We also assessed the potential synergistic effects of combining VaM with doxorubicin, a commonly used chemotherapeutic agent, in breast cancer cells. Understanding the molecular mechanisms of VaM-mediated anticancer effects and their potential as standalone or combination therapies can pave the way for developing more effective treatment strategies for patients with breast cancer.

## 2. Results

### 2.1. VaM Suppressed Cell Viability and Proliferation in Breast Cancer Cells

We investigated the effect of VaM on breast cancer cell viability and proliferation using the MTT assay. The viabilities of MCF-7, HCC-1428, and BT-474 cells were assessed in a dose-dependent manner. The results revealed that VaM significantly inhibited the viability of all three breast cancer cell lines in a dose-dependent manner (Figure 1A). Additionally, we performed a colony formation assay to evaluate the cell proliferative capacity. Notably, VaM treatment resulted in a significant reduction in colony formation compared to the untreated control group. These findings indicate that VaM effectively inhibits breast cancer cell viability and proliferation, highlighting its potential as a therapeutic agent against breast cancer.

### 2.2. VaM-Induced Apoptotic Cell Death in Breast Cancer Cells

To investigate the induction of apoptosis by VaM in breast cancer cells, we performed Annexin V/PI staining and western blot assays. Caspases play a crucial role in the apoptotic process [29]. Upon treatment with VaM, we observed a dose- and time-dependent increase in the cleaved form of caspase 3, accompanied by a decrease in the expression level of PARP, as analyzed via western blotting (Figure 2A,B). Additionally, using Annexin V/PI staining and flow cytometry, we observed a dose-dependent increase in the proportion of breast cancer cells undergoing early apoptosis. These results strongly support the ability of VaM to induce early apoptosis in breast cancer cells by activating caspase 3 and PARP, ultimately leading to cell death. The activation of these apoptotic pathways underscores the potential of VaM as an effective inducer of programmed cell death and a promising cytotoxic agent for breast cancer treatment.

### 2.3. VaM-Induced Cell Cycle Arrest in G1/S Phase in Breast Cancer Cells

Given that VaM regulates proliferation and apoptosis (Figure 2), we investigated its effect on the cell cycle of breast cancer cells. Western blotting (Figure 3A) revealed a dose-dependent decrease in the expression levels of key cell cycle regulators, including cyclin D1, cyclin E1, and CDK6, following treatment with VaM. Furthermore, flow cytometry provided additional evidence to support our findings (Figure 3B). VaM treatment reduced CDK6 (a cyclin-dependent kinase), cyclin D1, and cyclin E1 levels, which are crucial regulatory factors for the G1/S phase transition in the cell cycle. Collectively, our results demonstrate that VaM not only affects proliferation and apoptosis but also exerts a significant influence on the cell cycle of breast cancer cells.

### 2.4. VaM Modulated Src and STAT3 Phosphorylation in Breast Cancer Cells

Aberrant activation of the STAT3 signaling pathway plays a critical role in tumorigenesis and cancer progression [18]. To assess the effect of VaM on STAT3 in breast cancer, we evaluated the phosphorylation status of STAT3 and its upstream regulator, Src. Our findings revealed that VaM effectively inhibited the phosphorylation of both Src and STAT3 (Figure 4A). Additionally, we observed an upregulation of SHP-1, a negative regulator of STAT3 phosphorylation, upon VaM treatment. To further elucidate the mechanism of action, we conducted additional experiments using pervanadate, a potent inhibitor of protein tyrosine phosphatases. Interestingly, the observed increase in STAT3 phosphorylation with increasing pervanadate concentration, even in the presence of VaM, supported the inhibition of STAT3 tyrosine phosphatase activity by VaM (Figure 4B). Our findings demonstrated that VaM regulates Src and STAT3 through the involvement of SHP-1, leading to the induction of programmed cell death in breast cancer cells.

### 2.5. VaM Exhibited Potential Combined Effects with Doxorubicin in Breast Cancer Cells

Reducing the dosage of the anticancer drug doxorubicin to minimize its side effects is crucial for cancer treatment [30]. Hence, we investigated whether combining VaM and low-dose doxorubicin could maximize anticancer effects in breast cancer cells. Our results demonstrated that the combined treatment effectively reduced cell viability compared to treatment with doxorubicin or VaM alone, as observed in the MTT assay (Figure 5A). Additionally, protein expression analysis revealed that the combination treatment significantly decreased the phosphorylation of STAT3 and Src compared to the individual treatments while increasing the expression levels of cleaved-PARP and cleaved-caspase 3 (Figure 5B). In conclusion, our findings suggest that combining VaM and low-dose doxorubicin synergistically enhances the anticancer effects in breast cancer cells.

### 2.6. VaM Suppressed Tumor Growth by Regulating STAT3 Phosphorylation In Vivo

To investigate the in vivo effects of mistletoe extract (VaM), MCF-7 cells were xenografted and treated with intraperitoneal administration of VaM. The results revealed a dose-dependent reduction in tumor volume and weight in the VaM-treated group compared to the control group (NT). Furthermore, the doxorubicin and combination treatment groups exhibited significant decreases in tumor size and weight compared to the NT group, although no significant difference was observed between the doxorubicin alone and combination treatment groups (Figure 6A,C,D). High concentrations of VaM showed a slight decrease in body weight (Figure 6B). Immunoblotting analysis of tumor tissue demonstrated a significant decrease in STAT3 phosphorylation in the high-concentration VaM, doxorubicin alone, and combination treatment groups. Additionally, pro-caspase3 exhibited a slight decrease in the high-concentration VaM group, while a significant decrease was observed in the doxorubicin and combination treatment groups. These findings suggest that mistletoe extract induces apoptosis by modulating the STAT3 signaling pathway in an in vivo.

## 3. Discussion

This study aimed to investigate the potential anticancer effects of VaM on breast cancer cells and elucidate its underlying mechanisms. Our findings provide compelling evidence for the effectiveness of VaM in inhibiting breast cancer cell viability, proliferation, and apoptosis. Additionally, we demonstrated modulation of the cell cycle and key signaling pathways involved in breast cancer.

The inhibitory effect of VaM on cell viability was consistently observed in all three breast cancer cell lines. The significant reduction in colony formation further supported its anti-proliferative potential. These observations collectively highlight the ability of VaM to effectively inhibit breast cancer cell growth.

Apoptosis induction is a critical mechanism in targeted cancer therapy [31]. This study demonstrated that VaM effectively induces early apoptosis in breast cancer cells. This was evidenced by Annexin V/PI staining and flow cytometry, which revealed a dose-dependent increase in the proportion of apoptotic cells. Activation of caspase 3 and PARP cleavage further confirmed programmed cell death induction. These findings highlighted VaM as a potential therapeutic agent triggering breast cancer cell apoptosis.

Furthermore, we investigated the effects of VaM on cell cycle progression in breast cancer cells. Our results revealed a dose-dependent decrease in the expression of key cell cycle regulators, including Cyclin D1, Cyclin E1, and CDK6. These proteins are crucial for the G1/S phase transition, and their downregulation suggests potential VaM-induced cell cycle arrest in breast cancer cells. Disruption of cell cycle progression further contributes to the inhibitory effects of VaM on breast cancer cell growth.

The dysregulation of the STAT3 signaling pathway is frequently observed in breast cancer and contributes to tumor progression [18,32]. This study showed that VaM effectively inhibited the phosphorylation of both Src and STAT3. These findings suggest that VaM exerts its anticancer effects, at least in part, through the modulation of the STAT3 pathway. Upregulation of SHP-1, a negative regulator of STAT3 phosphorylation, further supports the inhibitory effect of VaM on STAT3 signaling. The inhibition of STAT3 phosphorylation and the subsequent activation of apoptotic markers, such as cleaved-PARP and cleaved-caspase 3, underline the involvement of the STAT3 pathway in the anticancer mechanisms of VaM.

An intriguing aspect of our study was the investigation of the combined effects of VaM and low-dose doxorubicin, a commonly used chemotherapeutic drug. Our results demonstrated a synergistic reduction in cell viability when VaM was combined with a low dose of doxorubicin. The combination treatment also led to a significant decrease in the phosphorylation of STAT3 and Src and an increase in apoptotic markers. These findings suggest a potential synergistic interaction between VaM and low-dose doxorubicin to induce apoptosis and inhibit key signaling pathways in breast cancer cells. This highlights the possibility of using combination therapy to maximize therapeutic efficacy while minimizing side effects.

Additionally, we investigated the in vivo effect of mistletoe extract (VaM) on breast cancer growth using a xenograft mouse model. Interestingly, we observed a significant reduction in tumor size and weight in the combination treatment group of mistletoe extract (VaM) and doxorubicin (Doxo) compared to the control group. However, there was no significant difference between the doxorubicin alone treatment group and the combination treatment group. These findings suggest that while VaM may be effective in breast cancer treatment, it may not enhance the therapeutic efficacy of doxorubicin in vivo as observed in in vitro experiments. Therefore, further investigation is needed to optimize the combination therapy. Body weight serves as an important indicator of overall health and well-being in animal studies. In our study, we observed a slight decrease in body weight at high concentrations of VaM. This finding highlights the need for caution when administering high doses of VaM to minimize potential side effects. To explore the underlying molecular mechanisms of the observed effects, we examined protein expression in tumor tissue. Immunoblotting analysis revealed a significant decrease in STAT3 phosphorylation in the high-concentration VaM, doxorubicin alone, and combination treatment groups. STAT3 plays a crucial role in cell proliferation, survival, and inflammation. The downregulation of STAT3 phosphorylation suggests that VaM may inhibit breast cancer growth by modulating the STAT3 signaling pathway. Furthermore, we observed a slight decrease in pro-caspase3 expression in the high-concentration VaM group and a significant decrease in the doxorubicin alone and combination treatment groups. Pro-caspase3 is a precursor of caspase3, a key executor of apoptosis. The reduction in pro-caspase3 expression indicates enhanced induction of apoptosis in the doxorubicin and combination treatment groups, potentially contributing to the observed inhibition of tumor growth.

## 4. Materials and Methods

### 4.1. Chemicals and Reagents

We obtained an extract of *Viscum album*, a parasitic plant that grows on *Malus domestica* (referred to as VaM) from Dalim BioTech (Wonju, Republic of Korea), which produces it as a product called Helixor M. Doxorubicin was purchased from Selleck Chem (Munich, Germany). Primary antibodies against STAT3, Phospho-STAT3 (Tyr705, Cat No. 9145s), Src, Phospho-Src, Phospho-Jak2, cleaved-caspase 3, and cleaved-PARP were purchased from Cell Signaling Technology (Beverly, MA, USA). Jak2, β-actin (Cat No. sc-47778), and secondary antibodies (Cat No. sc-516102 and sc-2004) were purchased from Santa Cruz Biotechnology (Dallas, TX, USA).

### 4.2. Cell Culture

MCF-7, BT-474, and HCC-1428 cells were purchased from the Korean Cell Line Bank (Seoul, Republic of Korea). All cell lines were cultured in RPMI-1640 medium containing 1% antibiotics and 10% fetal bovine serum at 37 °C and in a 5% CO_2_ incubator.

### 4.3. Cytotoxicity Assay

The cytotoxicity of VaM was measured using the 3-(4,5-dimethylthiazol-2-yl)-2,5-diphenyltetrazolium bromide (MTT) assay (Sigma Aldrich Co., St. Louis, MO, USA). MTT assay was performed as described previously [33]. Briefly, MCF-7 and BT-474 cells were seeded in 96-well plates (1 × 10^4^ cells/well) and treated with different concentrations of VaM (0, 62.5, 125, 250, 500, and 1000 μg/mL) for 24 or 48 h. After 24 h, the cells were exposed to the MTT solution until formazan was formed, and the absorbance was measured at 570 nm using a microplate reader (Bio-Rad, Hercules, CA, USA).

### 4.4. Colony Formation Assay

MCF-7, BT-474, and HCC-1428 cells treated with each concentration of VaM were seeded in 6-well plates (1 × 10^3^ cells/well) and cultured for 2 weeks at 37 °C and 5% CO_2_. After colony formation, the cells were fixed for staining using the Diff-Quick kit (Sysmex Corporation, Kobe, Hyogo, Japan).

### 4.5. Western Blotting Assay

Western blotting was performed as described previously [33,34,35]. To explain in detail, MCF-7, BT-474, and HCC-1428 cells (2 × 10^5^ cells/well) were treated with VaM and lysed in a lysis buffer (Cell Signaling Technology, Beverly, MA, USA). Proteins were separated by SDS-PAGE and transferred onto nitrocellulose membranes. Subsequently, the membranes were blocked with TBS + 0.1% Tween 20 containing skim milk for 1 h. The membranes were incubated with antibodies against cleaved-PARP (1:1000), cleaved-caspase 3 (1:1000), STAT3 (1:1000), Phospho-STAT3 (1:1000), Src (1:1000), Phospho-Src (1:1000), β-actin (1:1000), goat anti-mouse IgG-HRP (1:10,000), and goat anti-rabbit IgG-HRP (1:10,000). Protein expression was detected using ImageQuant LAS 500 (GE Healthcare Life Sciences, Sydney, Australia).

### 4.6. RNA Interference

The control and SHP-1 siRNAs were obtained from Bioneer (Daejeon, Republic of Korea). MCF-7 cells were seeded at a density of 7 × 10^4^ cells/well in 6-well plates and transfected with either SHP-1 siRNA or control siRNA using the INTERFERin^®^ transfection reagent (Polyplus-transfection SA, Illkirch, France). The siRNA mixture was then added to each well and incubated at 37 °C with 5% CO_2_ for 48 h. Subsequently, VaM treatment was administered for 24 h before cell harvesting.

### 4.7. Annexin V/PI Assay

MCF-7 cells were seeded at a density of 2 × 10^5^ cells per well and treated with VaM for 24 h. Following treatment, the cells were washed with 1× PBS and resuspended in 1× binding buffer containing FITC-tagged Annexin V and propidium iodide (PI) staining solution. Staining was performed at room temperature for 15 min. Apoptosis was analyzed using a FACSCanto II flow cytometer (BD Biosciences, Becton-Dickinson, Franklin Lakes, NJ, USA).

### 4.8. Animals

All animal procedures were conducted in accordance with the guidelines and were approved by the Institutional Animal Care and Use Committee of Kyung Hee University [KHUASP-23-017]. Four-week-old female athymic nu/nu mice were obtained from JA BIO Co. (Suwon-si, Republic of Korea). The mice were housed in a controlled environment with a 12 h light–dark cycle, constant temperature, and humidity. The mice were provided sterilized mouse chow and water ad libitum and allowed to acclimate for one week before the experiment. During this period, the mice were monitored for health and the absence of lesions, confirming their pathogen-free status.

### 4.9. Tumor Xenograft Experiments

Following tumor injection (5 × 10^6^ cells/mouse) on the right flank after one-week of stabilization, tumor diameters were measured three times a week. When the tumors reached a diameter of 60 mm^3^, the mice were randomly assigned to five treatment groups (*n* = 6/group): Group I (control) received PBS (100 μL, intraperitoneal injection, three times/week), Group II received VaM L (25 mg/kg, intraperitoneal injection, three times/week), Group III received VaM H (50 mg/kg, intraperitoneal injection, three times/week), Group IV received Doxorubicin (5 mg/kg, intraperitoneal injection, three times/week), and Group V received VaM H + Doxorubicin (VaM 50 mg/kg and Doxorubicin 5 mg/kg, intraperitoneal injection, three times/week). Treatment was administered for 8 days, starting at randomization (day 0). Tumor volumes and body weights were measured three times per week. After drug administration, the mice were euthanized, and the tumor tissues were collected and weighed. Subsequently, the tumor tissues were homogenized to extract proteins, and the protein expression levels were analyzed by western blotting.

### 4.10. Statistical Analysis

All experiments were repeated at least three times and expressed as mean ± standard deviation (SD). Significance was determined using the GraphPad Prism software (version 8.0; San Diego, CA, USA). Each comparison was performed using a one-way analysis of variance (ANOVA), followed by Dunnett’s test. Two-group comparisons were performed using unpaired *t*-tests.

## 5. Conclusions

In conclusion, our study provides evidence for the potential therapeutic benefits of VaM in breast cancer treatment. VaM effectively inhibits cancer cell viability and proliferation, induces apoptosis, and regulates cell cycle progression through the STAT3 signaling pathway. The combination of in vitro and in vivo results strengthens the validity of our findings and highlights the potential of VaM as a promising therapeutic agent for breast cancer.

## Figures and Tables

**Figure 1 ijms-24-11988-f001:**
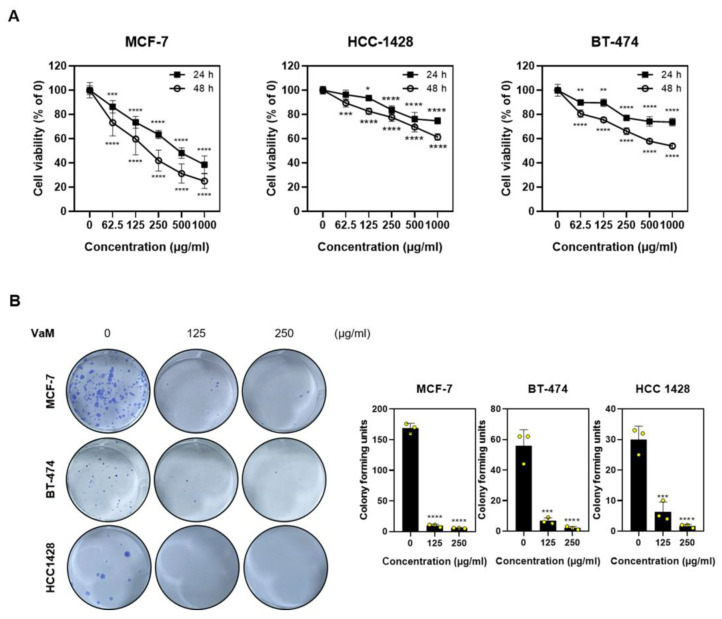
Inhibitory effects on the viability and proliferation of breast cancer cells. (**A**) The cells were treated with VaM for 24 or 48 h, and cell viability was assessed using the MTT assay. (**B**) Colony formation assay was performed to evaluate the inhibitory effect of VaM on cell proliferation. Cells treated with or without VaM were reseeded, and after 14 days, the colonies were stained and counted. The graph shows the quantification of stained colonies relative to that of the untreated group. The data were quantified based on an untreated group of each cell. * *p* < 0.05, ** *p* < 0.01, *** *p* < 0.001, and **** *p* < 0.0001 compared to the condition with zero treatment concentration. The results are presented in the bar chart as the mean ± SD of the three independent experiments. Yellow circles represent the results of three independent experiments.

**Figure 2 ijms-24-11988-f002:**
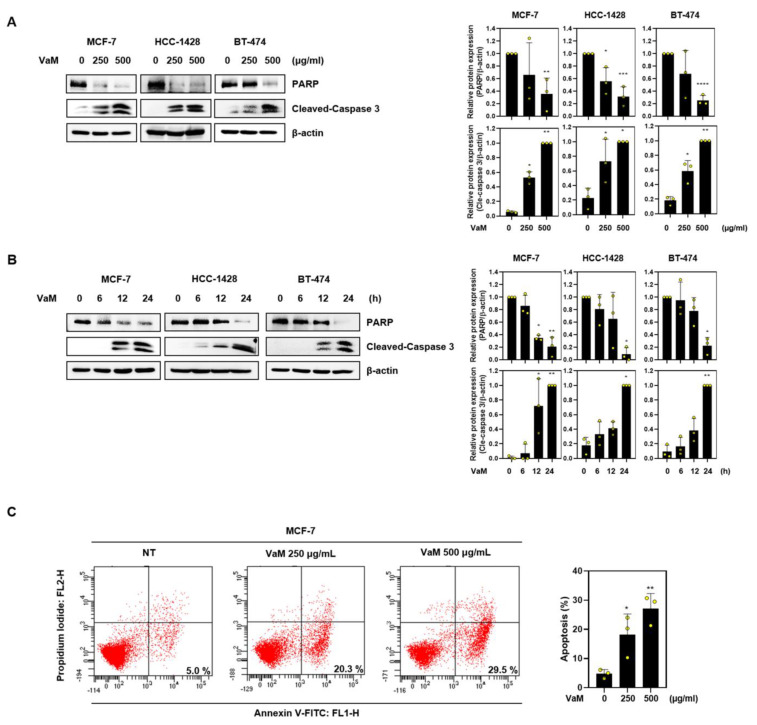
Apoptotic effect of VaM on breast cancer cells. Cells were treated with VaM (**A**) at various concentrations for 24 h and (**B**) 500 μg/mL at the indicated time points. The protein expression of Pro-PARP and cleaved-caspase 3, which are apoptosis markers, was confirmed by immunoblotting. The data were quantified and compared to those of the untreated groups. (**C**) Flow cytometry was used to analyze Annexin V/PI-stained cells. * *p* < 0.05, ** *p* < 0.01, *** *p* < 0.001, and **** *p* < 0.0001 compared to the condition with zero treatment concentration. The results are presented in the bar chart as the mean ± SD of the three independent experiments. Yellow circles represent the results of three independent experiments.

**Figure 3 ijms-24-11988-f003:**
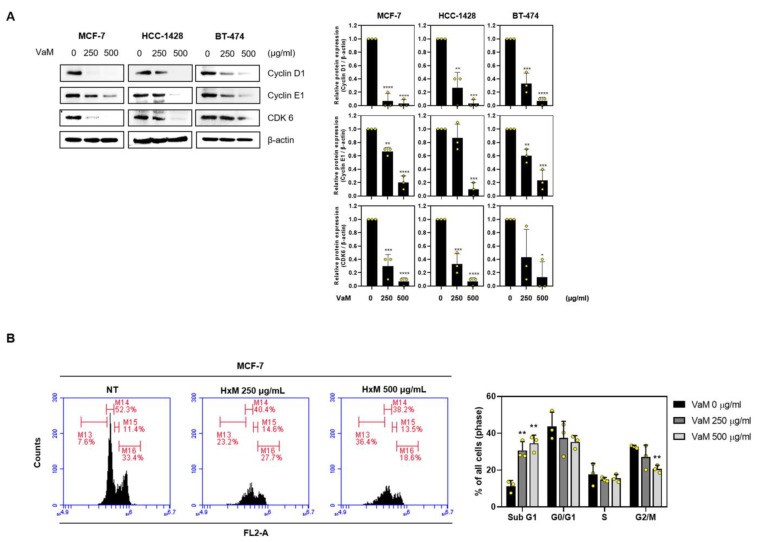
VaM -induced cell cycle arrest in breast cancer cells. (**A**) The cells were treated with VaM at the indicated concentrations for 24 h. The protein expression of cyclin D1, cyclin E1, and CDK6, which are cell cycle-associated markers, was confirmed by immunoblotting. The data were quantified and compared to that of the untreated groups. (**B**) Flow cytometry was used to analyze PI-stained cells. * *p* < 0.05, ** *p* < 0.01, *** *p* < 0.001, and **** *p* < 0.0001 compared to the condition with zero treatment concentration. The results are presented in the bar chart as the mean ± SD of the three independent experiments. Yellow circles represent the results of three independent experiments.

**Figure 4 ijms-24-11988-f004:**
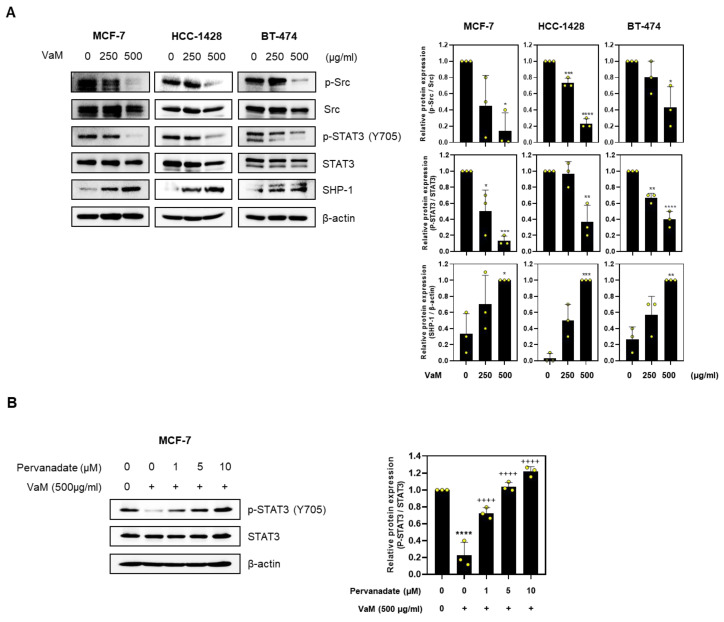
Modulation of Src and STAT3 phosphorylation in breast cancer cells by VaM. (**A**) The protein expression of p-Src, Src, p-STAT3 (Y705), STAT3 (Y705), and SHP-1 was confirmed by immunoblotting. The data were quantified and compared to that of the untreated groups. (**B**) MCF-7 cells pre-treated with pervanadate were treated with VaM, and the protein expression levels of p-STAT3 and STAT3 were examined by immunoblotting. * *p* < 0.05, ** *p* < 0.01, *** *p* < 0.001, **** *p* < 0.0001 compared to the no treatment concentration group. ^++++^
*p* < 0.0001 compared to VaM (500 μg/mL) group. The results are presented in the bar chart as the mean ± SD of the three independent experiments. Yellow circles represent the results of three independent experiments.

**Figure 5 ijms-24-11988-f005:**
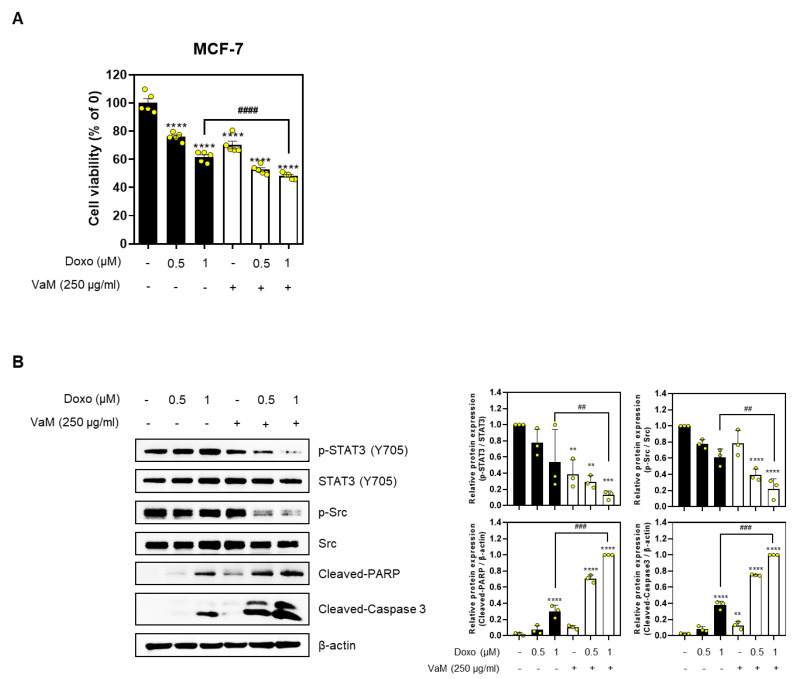
Potential combined effects of doxorubicin and VaM in breast cancer cells. Doxo was treated with or without VaM (250 μg/mL) for 24 h in MCF-7 cells. (**A**) Cell viability was determined using MTT assay. (**B**) The protein expression of p-STAT3 (Y705), STAT3, p-Src, Src, cleaved-PARP, and cleaved-caspase 3 was confirmed by immunoblotting. ** *p* < 0.01, *** *p* < 0.001, **** *p* < 0.0001 compared to the no treatment concentration group. ^##^
*p* < 0.01, ^###^
*p* < 0.001, and ^####^
*p* < 0.0001 compared to only Doxo (1 μM) group. The results are presented in the bar chart as the mean ± SD of the three independent experiments. Yellow circles represent the results of three independent experiments.

**Figure 6 ijms-24-11988-f006:**
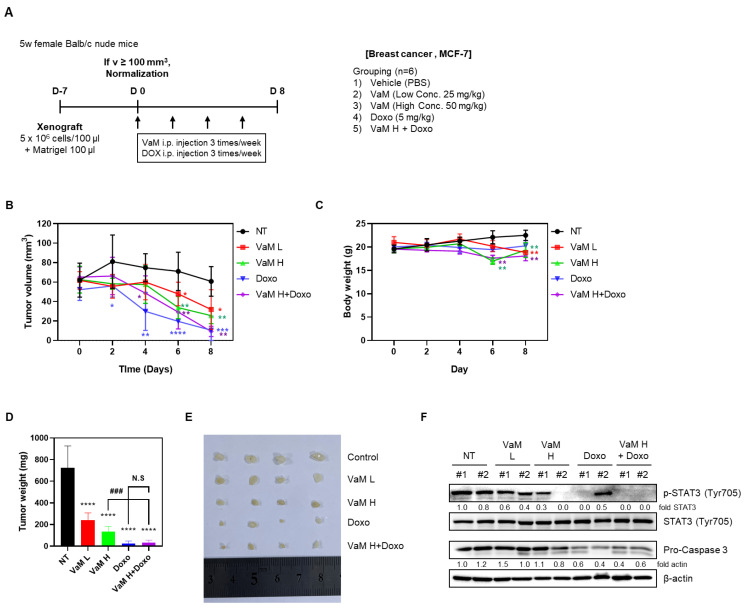
Effects of VaM on breast cancer cell growth in a xenograft mouse model. (**A**) Schematic representation of the experimental methods described in the Materials and Methods section. MCF-7 cells (5 × 10^6^ cells/mouse) were mixed with Matrigel (1:1, *v*/*v*) and injected subcutaneously. Mice were randomly assigned to 6 groups and intraperitoneally administered 100 μL of PBS or each drug (25 and 50 mg/kg of VaM and 20 mg/kg of doxorubicin) 3 times a week for 14 days. The tumor volume was measured, and the mice were euthanized at the end of the treatment. (**B**) Changes in body weight and (**C**) tumor volume were monitored for 2 weeks after drug administration. (**D**) Representative photographs of harvested tumors from each treatment group. (**E**) Tumor weight was measured. (**F**) Immunoblotting analysis of tumor tissue lysates for p-STAT3, STAT3, cleaved-PARP, and pro-caspase 3 protein expression levels. * *p* < 0.05, ** *p* < 0.01, *** *p* < 0.001, **** *p* < 0.0001 compared to the no treatment concentration group. ^###^
*p* < 0.001 compared to VaM H group. N.S means no significant difference.

## Data Availability

All data presented in this study are available from the corresponding author upon reasonable request.

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
