# Peer review of "Viscum album Induces Apoptosis by Regulating STAT3 Signaling Pathway in Breast Cancer Cells"

_ijms, 2023, doi:10.3390/ijms241511988_

Round 1

Reviewer 1 Report

In this manuscript, the authors explore the potential effects and molecular mechanism of VaM in breast cancer cells. The authors discovered that VaM significantly inhibited cell viability and proliferation and induced apoptosis in a dose-dependent manner. Next the authors also discovered that VaM regulated cell cycle and effectively inhibited activation of the STAT3 signaling pathway through SHP-1. Finally, the authors discovered that VaM inhibited tumor growth and modulated key molecular markers associated with breast cancer progression in vivo. In general, the authors present lots of results, and my assessment of this manuscript is positive. However, I think some minor issues should be addressed prior to publication in IJMS.

Specific comments:

1.    In figure 3, the figure legend does not match figure

2.    In figure 6F, please quantify the protein expression of p-STAT3 AND Pro-Caspase 3.

Author Response

Dear Chief Editor

Resubmission of manuscript reference ID: ijms-2518452

We would like to thank you for the letter dated 24th July, 2023, and we hereby resubmit a revised copy of the manuscript for your consideration. We would also like to express our thanks to you for insightful comments that have greatly helped improve the quality of the manuscript.

In this manuscript, we have studied carefully and revised and added details, reflecting the overall advice of previous reviewers. We hope that we addressed well to comments and also wish our revised manuscript would be accepted for publication in International Journal of Molecular Sciences.

Best regards,

Hyeung-Jin Jang

Reviewer 1

In this manuscript, the authors explore the potential effects and molecular mechanism of VaM in breast cancer cells. The authors discovered that VaM significantly inhibited cell viability and proliferation and induced apoptosis in a dose-dependent manner. Next the authors also discovered that VaM regulated cell cycle and effectively inhibited activation of the STAT3 signaling pathway through SHP-1. Finally, the authors discovered that VaM inhibited tumor growth and modulated key molecular markers associated with breast cancer progression in vivo. In general, the authors present lots of results, and my assessment of this manuscript is positive. However, I think some minor issues should be addressed prior to publication in IJMS.

Specific comments:

  1. In figure 3, the figure legend does not match figure

(Response) Thanks for your detailed comments. The figure legend was matched appropriately.

  1. In figure 6F, please quantify the protein expression of p-STAT3 AND Pro-Caspase 3.

(Response) As the reviewer pointed out, we have already quantified each fold value under the protein expression image.

Reviewer 2 Report

In a study "Viscum album induces Apoptotic Cell Death by Regulating
STAT3 Signaling Pathway in Breast Cancer Cells" by Ye-Rin Park et al, the Authors investigate the potential anticancer effects of Viscum album, a parasitic plant on breast cancer cells and try to explor the underlying mechanisms. The manuscript is relatively well written, and the results are presented clearly (besides specific comment 2). The conclusions are supported by the results.

Specific comments:

1. Title - please change "apoptotic cell death" to just "apoptosis".

2. Certain panels of figures can be shown enlarged e.g., Fig. 1B (graphs). In general, the Figures should be enlarged as they are barely visible.

3. Fig. 2 - (1) please change pro-PARP/PARP to "full-length PARP". (2) cleaved PARP (~86 kDa) should be shown on Western blots to indicate that cell death is induced via apoptosis.

4. results 3.3 - the title is not reflecting the results. The subG1 fraction is only increased, which is probably associated with induction of apoptosis.

5. It is not clear why certain experiment were performed using a single cell line e.g., Fig. 2C, Fig. 3B.

acceptable

Author Response

Dear Chief Editor

Resubmission of manuscript reference ID: ijms-2518452

We would like to thank you for the letter dated 24th July, 2023, and we hereby resubmit a revised copy of the manuscript for your consideration. We would also like to express our thanks to you for insightful comments that have greatly helped improve the quality of the manuscript.

In this manuscript, we have studied carefully and revised and added details, reflecting the overall advice of previous reviewers. We hope that we addressed well to comments and also wish our revised manuscript would be accepted for publication in International Journal of Molecular Sciences.

Best regards,

Hyeung-Jin Jang

Reviewer 2

In a study "Viscum album induces Apoptotic Cell Death by Regulating
STAT3 Signaling Pathway in Breast Cancer Cells" by Ye-Rin Park et al, the Authors investigate the potential anticancer effects of Viscum album, a parasitic plant on breast cancer cells and try to explor the underlying mechanisms. The manuscript is relatively well written, and the results are presented clearly (besides specific comment 2). The conclusions are supported by the results.

Specific comments:

  1. Title - please change "apoptotic cell death" to just "apoptosis".

(Response) Thanks for your detailed comments. We revised the title.

  1. Certain panels of figures can be shown enlarged e.g., Fig. 1B (graphs). In general, the Figures should be enlarged as they are barely visible.

(Response) As mentioned above, the size of the graph has been increased.

  1. Fig. 2 - (1) please change pro-PARP/PARP to "full-length PARP". (2) cleaved PARP (~86 kDa) should be shown on Western blots to indicate that cell death is induced via apoptosis.

(Response) Based on Raw data page 2, it can be observed that the cleaved form of PARP was detected. However, when we performed the experiment three or more times to obtain this data, the cleaved form was not consistently detectable. Therefore, we decided to present only the pro-PARP form in the figure.

  1. results 3.3 - the title is not reflecting the results. The subG1 fraction is only increased, which is probably associated with induction of apoptosis.

(Response) G1/S phase arrest refers to a state where the progression from the G1 phase to the S phase of the cell cycle is halted.

Based on Figure 3B, when cells were treated with varying concentrations of the drug, the cell percentage in S phase and G2/M phase decreased, indicating an arrest in the S phase. Additionally, the cell cycle arrest at the G1/S phase led to an accumulation of cells in the Sub G1 phase due to cell death (apoptosis).

To further investigate and confirm this phenomenon, protein expression was examined, as shown in Figure 3A. The protein expression levels of Cyclin D1 and its binding partner, CDK6, which regulate the progression of the G1/S phase, significantly decreased after treating the cells with the drug at different concentrations. Moreover, after Cyclin D1 and CDK6, which play a role in controlling the cell cycle, form complexes to regulate the cell cycle, Cyclin E1 is activated to prepare for the progression to the S phase. Notably, the protein level of Cyclin E1 also decreased upon drug treatment.

Collectively, these findings provide clear evidence that the drug VaM induces G1/S phase arrest, subsequently leading to cell death.

  1. It is not clear why certain experiment were performed using a single cell line e.g., Fig. 2C, Fig. 3B.

(Response) When VaM was treated, MCF-7, which had the lowest cell viability rate and the best efficiency in cell cycle related protein expression, was representatively investigated.

Round 2

Reviewer 2 Report

The comments have been adequately addressed.